# Targeted COVID-19 Vaccination (TAV-COVID) Considering Limited Vaccination Capacities—An Agent-Based Modeling Evaluation

**DOI:** 10.3390/vaccines9050434

**Published:** 2021-04-27

**Authors:** Beate Jahn, Gaby Sroczynski, Martin Bicher, Claire Rippinger, Nikolai Mühlberger, Júlia Santamaria, Christoph Urach, Michael Schomaker, Igor Stojkov, Daniela Schmid, Günter Weiss, Ursula Wiedermann, Monika Redlberger-Fritz, Christiane Druml, Mirjam Kretzschmar, Maria Paulke-Korinek, Herwig Ostermann, Caroline Czasch, Gottfried Endel, Wolfgang Bock, Nikolas Popper, Uwe Siebert

**Affiliations:** 1Department of Public Health, Health Services Research and Health Technology Assessment, Institute of Public Health, Medical Decision Making and Health Technology Assessment, UMIT—University for Health Sciences, Medical Informatics and Technology, Eduard-Wallnoefer-Zentrum 1, A-6060 Hall in Tirol, Austria; beate.jahn@umit.at (B.J.); gaby.sroczynski@umit.at (G.S.); nikolai.muehlberger@umit.at (N.M.); julia.santamaria-navarro@umit.at (J.S.); michael.schomaker@umit.at (M.S.); igor.stojkov@umit.at (I.S.); 2dwh GmbH, dwh Simulation Services, Neustiftgasse 57–59, A-1070 Vienna, Austria; martin.bicher@tuwien.ac.at (M.B.); claire.rippinger@dwh.at (C.R.); christoph.urach@dwh.at (C.U.); nikolas.popper@tuwien.ac.at (N.P.); 3Institute of Information Systems Engineering, TU Wien, Favoritenstraße 11, A-1050 Vienna, Austria; 4Center for Infectious Disease Epidemiology and Research, University of Cape Town, Barnard Fuller Building, Anzio Rd, Observatory, Cape Town 7935, South Africa; 5Division for Quantitative Methods in Public Health and Health Services Research, Department of Public Health, Health Services Research and Health Technology Assessment, UMIT—University for Health Sciences, Medical Informatics and Technology, Eduard-Wallnoefer-Zentrum 1, A-6060 Hall in Tirol, Austria; daniela.schmid@umit.at; 6Department of Internal Medicine II, Medical University of Innsbruck, Anichstraße 35, 6020 Innsbruck, Austria; guenter.weiss@i-med.ac.at; 7Center of Pathophysiology, Infectiology & Immunology (OEL), Institute of Specific Prophylaxis and Tropical Medicine, Medical University of Vienna, Kinderspitalgasse 15, 1090 Vienna, Austria; ursula.wiedermann-schmidt@meduniwein.ac.at; 8Center of Virology, Medical University of Vienna, Kinderspitalgasse 15, 1090 Vienna, Austria; monika.redlberger@meduniwien.ac.at; 9UNESCO Chair on Bioethics, Medical University of Vienna, Waehringerstrasse 25, 1090 Vienna, Austria; christiane.druml@meduniwien.ac.at; 10Julius Center for Health Sciences and Primary Care, University Medical Center Utrecht, Heidelberglaan 100, 3584CX Utrecht, The Netherlands; m.e.e.kretzschmar@umcutrecht.nl; 11Ministry of Social Affairs, Health, Care and Consumer Protection, Stubenring 1, 1010 Vienna, Austria; Maria.Paulke-Korinek@gesundheitsministerium.gv.at; 12Austrian National Public Health Institute/Gesundheit Österreich GmbH, Stubenring 6, 1010 Vienna, Austria; herwig.ostermann@goeg.at (H.O.); caroline.czasch@goeg.at (C.C.); 13Austrian Federation of Social Insurances, Kundmanngasse 21, 1030 Vienna, Austria; Gottfried.Endel@sozialversicherung.at; 14Department of Mathematics, TU Kaiserslautern, Gottlieb-Daimler-Straße 48, 67663 Kaiserslautern, Germany; bock@mathematik.uni-kl.de; 15Association for Decision Support for Health Policy and Planning, DEXHELPP, Neustiftgasse 57–59, A-1070 Vienna, Austria; 16Institute for Technology Assessment and Department of Radiology, Massachusetts General Hospital, Harvard Medical School, 101 Merrimac St., Boston, MA 02114, USA; 17Center for Health Decision Science, Departments of Epidemiology and Health Policy & Management, Harvard T.H. Chan School of Public Health, 718 Huntington Avenue, Boston, MA 02115, USA

**Keywords:** SARS-CoV-2, COVID-19, vaccination, prioritization, vaccination strategy, optimization, decision-analytic modeling, agent-based simulation, health policy decision making, policy guidance

## Abstract

(1) Background: The Austrian supply of COVID-19 vaccine is limited for now. We aim to provide evidence-based guidance to the authorities in order to minimize COVID-19-related hospitalizations and deaths in Austria. (2) Methods: We used a dynamic agent-based population model to compare different vaccination strategies targeted to the elderly (65 ≥ years), middle aged (45–64 years), younger (15–44 years), vulnerable (risk of severe disease due to comorbidities), and healthcare workers (HCW). First, outcomes were optimized for an initially available vaccine batch for 200,000 individuals. Second, stepwise optimization was performed deriving a prioritization sequence for 2.45 million individuals, maximizing the reduction in total hospitalizations and deaths compared to no vaccination. We considered sterilizing and non-sterilizing immunity, assuming a 70% effectiveness. (3) Results: Maximum reduction of hospitalizations and deaths was achieved by starting vaccination with the elderly and vulnerable followed by middle-aged, HCW, and younger individuals. Optimizations for vaccinating 2.45 million individuals yielded the same prioritization and avoided approximately one third of deaths and hospitalizations. Starting vaccination with HCW leads to slightly smaller reductions but maximizes occupational safety. (4) Conclusion: To minimize COVID-19-related hospitalizations and deaths, our study shows that elderly and vulnerable persons should be prioritized for vaccination until further vaccines are available.

## 1. Introduction

Worldwide, the newly emerged pandemic severe acute respiratory syndrome coronavirus type-2 virus (SARS-CoV-2) has led to enormous health, social, and economic burdens. Severe cases of coronavirus disease 2019 (COVID-19) are associated with increased mortality and hospitalizations, overwhelming many healthcare systems. In the absence of vaccines, many countries have implemented non-pharmaceutical interventions (NPI) to control and slow down the epidemic spread (e.g., social distancing measures, mandatory use of face masks, travel restrictions, contact tracing, and mass testing) [1,2,3,4].

Today, 69 vaccine candidates are being tested in clinical trials, and the first vaccines received market authorization approvals starting in late 2020 [5,6]. In general, vaccines can provide direct protection, for example, prevention of disease, or they can prevent infections and thereby prevent transmission too. Depending on these characteristics, the vaccination strategy may focus on the protection of those who are at the highest risk of severe disease such as patients with cancer, human immunodeficiency viruses (HIV), pregnant women, or patients undergoing immunosuppression treatments [7,8,9]. On the other hand, the strategy could target those who are spreading the disease, or it could consider both aspects. Currently, vaccines show promising results for the protection of severe diseases. However, the influence of the vaccines on viral load and reduction of transmission (achieved sterilizing immunity) is not yet fully known [10,11,12,13]. Elderly and vulnerable individuals with comorbidities are at greatest risk for severe COVID-19 and thus would directly benefit the most from vaccination independent of virus carriage or reduction of transmission [7,8,9,14]. Vaccination of other specific subpopulations (e.g., children or young adults who tend to have more contacts and are more mobile), however, may be an optimal target group to successfully prevent virus transmission and epidemic spread of the virus, as has been reported for influenza epidemics, although children appear to play a less important role for of SARS-CoV2 as compared to influenza viruses [15]. Moreover, vaccinating essential workers including healthcare workers (HCW) is crucial to maintain healthcare and other system services and reduces the risk of virus transmission to patients and nursing home residents [16,17]. 

As such, systematically investigating the tradeoffs between these strategies is essential to provide evidence-based public health guidance for health policy decision making on vaccine distribution, particularly in the initial phases when vaccine availability is limited. Decision makers need to wisely target limited vaccine capacities for subpopulations (e.g., risk groups, HCW, and settings with elevated transmission rates) to optimize overall health and non-health outcomes.

In Austria, considerations for the initial phase of a potential population-based SARS-CoV-2 vaccination also included a scenario in which HCW are vaccinated first [18]. The Austrian government has ordered a total of 30.5 million doses from different pharmaceutical companies in the EU joint procurement for COVID-19 vaccines with the aim of providing the needed vaccines for the total Austrian population. The BioNTech/Pfizer vaccine BNT162b2 (Comirnaty, Biontech Manufacturing GmbH, Mainz, Deutschland; Pfizer, New York, NY, USA) achieved marketing authorization in the EU on 21 December 2020, the Moderna vaccine mRNA-1273 (Moderna, Cambridge, MA, USA) on 6 January 2021, and the AstraZeneca vaccine ChAdOx1-S (AstraZeneca, Cambridge, UK) on 29 January 2021. In the first quarter of 2021, in total about 2 million doses are expected to be available in Austria; in the second quarter, 7.2 million doses are expected. These numbers may increase if further marketing authorizations for other vaccines are granted. However, which vaccines and how many doses will be available with market authorization approval for the initial phase as of January 2021 was still unclear at the time of our study and changed repeatedly. It was, however, already known that the vaccine supply in Austria would be limited, and prioritization would be required. Additionally, it is also not known how many people would elect to receive a SARS-CoV-2 vaccine, and how effective the first available vaccines would be for different age-vulnerable patient groups. Despite of all these uncertainties, fast decisions must be made regarding the vaccine allocation strategy. In Austria, scientific experts, including public health experts and decision-analytic modelers, advise the government on the impact of different public health options. 

In this decision-analytic study, we used a stepwise-model-informed approach based on sequential optimization to quantify the impact of different SARS-CoV-2 vaccination strategies on cumulative COVID-19-related mortality and incidence of hospitalizations as prioritization criteria in Austria. This study, supported in part by the Society for Medical Decision Making (SMDM) COVID-19 Decision Modeling Initiative, was explicitly designed to inform public health policy decision makers at an early point in time regarding the optimal distribution sequence of SARS-CoV-2 vaccination (once available) according to specific benefit criteria, accounting for limited vaccination capacities and adherence in order to support evidence-based and outcome-oriented vaccination prioritization in combination with further containment measures in Austria.

## 2. Materials and Methods

We used decision-analytic modeling to compare multiple sequential prioritization rules targeting specific subgroups to identify strategies minimizing deaths and hospitalizations over an analytic time horizon of 6 months after availability of the first vaccine doses in Austria [19]. To consider the simultaneous impact of hospitalizations, mortality and spread over time, we applied a previously published agent-based population model that is currently used to inform Austrian healthcare decision-making bodies [20,21,22,23,24]. We followed international guidelines of the ISPOR-SMDM Joint Modeling Good Research Practices Task Force for the development and analysis of our model, as well as for the reporting of our methods and results [25,26,27].

As this project was designed to guide the decision-making authorities in Austria, we established a Standing Policy and Expert Panel TAV-COVID (SPEP TAV-COVID), including 13 national and international experts from different disciplines and institutions. Experts met in two expert workshops and provided continuous advice for interdisciplinary and discipline-specific questions throughout the project.

### 2.1. Agent-Based Simulation Model

Briefly, in our model we consider the Austrian population represented by about nine million statistical representatives (”agents”) with a contact network accounting for different locations, such as households, workplaces, schools, and leisure time [20]. Characteristics and health states of individuals are tracked during the analysis on an individual level (micro-simulation). The pathway of virus transmission and COVID-19 disease of individuals is described by potential health states and events starting with healthy individuals who can get infected via contact with infected individuals. In the model, infected individuals can be either detected based on specific COVID-19-related symptoms or testing, or they remain undetected and could become infectious. For detected infected individuals, we distinguish between the time period from infection to symptom onset and a “notification delay” that describes the time between symptom onset, testing, positive test result, and the time at which the COVID-19 patient is recorded as a confirmed case in the official surveillance system. Confirmed cases are categorized into different disease severity states: mild cases recovering at home, severe cases treated in hospital, and critical cases requiring treatment in an intensive care unit (ICU). Disease severity is assumed to be dependent on age and comorbidities [28] (see Appendix A
Table A3 and Table A4). 

General parameters of the model are described in Bicher et al. [18], and parameters specific to the implementation of vaccination are summarized in the Appendix A. For all simulations, it was assumed that other protective measures against SARS-CoV-2 spread already implemented before the start of vaccination are maintained during the 6 month analytic time horizon of the analysis. The probability of an infection occurring during a single contact between an infectious and a susceptible individual has been determined by calibrating the model such that the model results match the number of detected cases of COVID-19 during the initial exponential growth in Austria in March 2020 [21]. 

### 2.2. Vaccine Effectiveness, Sterilizing Effect, and Vaccination Participation Rate

In all simulations, we conservatively assumed a vaccination effectiveness of 70% for individuals younger than 65 years and a reduced effectiveness of 60% for individuals 65 years of age and older. This assumption on vaccine effectiveness was based on expert opinions considering early trial results, the expectation that various vaccines with different effectiveness levels would be approved, and the fact that effectiveness in real-world applications might be somewhat lower than in the trials due to the selected patient population. Vaccines may provide sterilizing immunity, which is effectively preventing infection of vaccinated individuals but does not prevent viral transmissions to others. Alternatively, vaccines may induce non-sterilizing immunity, which is effectively preventing (severe) disease but not asymptomatic infections and transmissions to others. For vaccines inducing sterilizing immunity, we assumed that 70% of the vaccinated persons are protected against infection (60% for individuals 65 years of age). For vaccines inducing non-sterilizing immunity, vaccinated persons are protected against disease; that is, they either develop mild symptoms or remain asymptomatic with a ratio of cases mild: asymptomatic of 2:7. In younger age groups, the number of severe cases was reduced proportionally; that is, we applied the same relative reduction of severe cases as in the vaccinated elderly. Participation rates in the vaccination program were modeled depending on population groups (see Appendix A
Table A1).

### 2.3. Vaccination Target Groups and Prioritization Strategies

To evaluate the effect of different sequential strategies for the vaccination of different target groups, we considered five a priori-defined population groups, which were defined in a workshop with the Standing Policy and Expert Panel TAV-COVID: (1) elderly: 65 years and older (E; 1.7 million); (2) middle aged: 45–64 years old (M; 2.7 million); (3) young adults: 15–44 years old (Y; 3.4 million) [29]; (4) vulnerable: individuals with comorbidities leading to an increased risk for a severe course of COVID-19 (V; 2.9 million) [30,31]; and (5) HCW: including medical staff and caregivers in hospitals and outpatient care, medical practitioners, mobile care, long-term care facilities, care for the disabled, etc., totaling around 218,000 HCW (the age structure of these groups can be found in Appendix A
Table A2). Vulnerable individuals include individuals with pre-existing comorbidities leading to an increased risk of a severe course of COVID-19 disease in case of infection [28]. Increased risks were expressed as age-specific and comorbidity-specific odds ratios (OR) (diabetes (OR: 2.04), chronic kidney disease (OR: 2.23), chronic heart disease (OR: 3.50), chronic respiratory disease (OR: 2.11), chronic liver disease, (OR: 1.29), cancer (OR: 2.20)), and hypertension (OR: 2.83); for prevalence and absolute size of risk factor groups see Appendix A
Table A3 and Table A4).

### 2.4. Base-Case, Scenario and Sensitivity Analyses

In addition to the primary optimization criteria deaths and hospitalizations (ICU, non-ICU), we evaluated new confirmed incident COVID-19 cases. 

In order to optimize health outcomes along an increasing availability of vaccine doses, we used a stepwise approach based on sequential decision analysis to assess vaccine effects in different target groups. Particularly, the stepwise approach considered two facts: first, after having vaccinated some eligible and participating individuals of one target group, the next best target group must be identified, and so on, to generate a sequential prioritization list; second, the fact that the immunity situation changes over time with more and more vaccinated (and immune) individuals. Therefore, after identifying the optimal target group for the first batch of vaccine doses, the optimal vaccination allocation must be assessed de novo in a stepwise manner for the next batches conditional on increased immunity in the subgroups and overall population.

At the start of the modeling time frame, we assumed an initially limited number of vaccine doses, sufficient to fully vaccinate 200,000 individuals. We chose this group size (batch size) because this was the number of first batches expected to be received in Austria and because it allowed us to differentiate sequential effects on relatively small subgroups during the very first phase of vaccination. In the simulation, the spread of disease in the population is monitored over 6 months after vaccination, and the relative risk reduction (RRR) was calculated for each strategy and the evaluated outcomes are reported compared to no vaccination. We also assumed that protective immunity will last for at least 6 months after vaccination.

Next, we performed a stepwise optimization analysis to derive a prioritization sequence to guide vaccine administration to the 2.45 million individuals next in line after the initial 200,000. We performed two independent optimization analyses. First, we aimed to minimize deaths (primary analysis), and second, we minimized hospitalizations (secondary analysis). In the simulation, 2.45 million individuals received vaccines using batches for 200,000, 750,000, 750,000, and 750,000 individuals in a stepwise fashion. For the optimization of the first batch, we performed five separate simulations comparing the effect of providing the batch of 200,000 to either of the five target groups (TG) and compared RRR for deaths or hospitalizations in the entire population. The target group leading to the maximum RRR was chosen to receive the first batch (TG1_200,000_). The next step started with the 200,000 individuals identified by TG1_200,000_, and the next batch of 750,000 vaccines was distributed to yet unvaccinated target group members in separate simulations. Now, the target group achieving the maximum RRR for deaths or hospitalizations was selected as the target group to receive the second batch (TG2_750,000_). This stepwise procedure was repeated to determine TG3_750,000_ and TG4_750,000_ to determine the optimal vaccination sequence for all of the 2.45 million individuals. 

In addition, we simulated scenarios in which the first vaccines were assigned by policy rule to 200,000 healthcare workers. In this scenario analysis, we derived the subsequent prioritization sequence for further groups conditional on starting with healthcare workers. 

All simulations were performed for scenarios with non-sterilizing and sterilizing vaccines. In sensitivity analyses, we evaluated the prioritization sequence for a non-sterilizing vaccine with 100% effectiveness. Reported results include total and stepwise RRRs of selected strategies with respect to all evaluated outcomes. For all simulations, we report an average of five simulation runs to reduce random error.

### 2.5. Model Validation

The agent-based model was validated internally and externally on several levels [32]: (1) face validity (i.e., by clinical experts and modeling experts), (2) internal validation (e.g., debugging, consistency, and plausibility checks), and (3) external validation with results from published literature [21].

### 2.6. Standing Policy and Expert Panel

The project team was advised by members of the Standing Policy and Expert Panel (SPEP TAV-COVID) and other national and international experts. Advice included the selection of target groups, analytic time-horizon, stepwise strategies, batch sizes, optimization criteria (outcomes), model structure, appropriateness of identified data and sources, parameter transformation for use in the models, analytic methods, and interpretation and communication of results and limitations, as well as the reporting needs for Austrian policy and decision makers. The SPEP consists of (13) experts with experience from Austrian healthcare organizations and decision-making authorities, and further national and international experts in public health, epidemiology, virology, infectious disease medicine, vaccination, and ethics.

## 3. Results

Results are reported separately for the base-case analysis of batches with vaccination doses for 200,000 individuals and for 2.45 million individuals. Further scenarios and results are provided in the Appendix A.

### 3.1. Vaccination of the First 200,000 Individuals 

In both scenarios (i.e., non-sterilizing and sterilizing vaccines) for distributing doses to vaccinate the first 200,000 individuals, the maximum relative reduction in deaths and hospitalizations compared to no vaccination can be achieved by vaccination of individuals at age 65 or older first followed by vulnerable individuals with comorbidities with an increased risk for a severe course of disease once infected. Starting vaccinations with the age group 65+ years leads to a 3- to 5-fold relative reduction in deaths and hospitalizations than starting with HCW, depending on the type of vaccine and outcome considered. 

The population level impact of vaccinating the first 200,000 individuals in different target groups is displayed in Figure 1 for vaccines inducing non-sterilizing and sterilizing immunity, respectively.

### 3.2. Vaccination of the First 2.45 Million Individuals 

The analysis based on the stepwise optimizations for a vaccination of 2.45 million individuals yielded the same prioritization sequence as found in the evaluation of the first batch only, that is, a prioritization of individuals age 65 or older followed by vulnerable individuals. If the first vaccinations are assigned to HCW, the stepwise optimizations yielded a subsequent prioritization for individuals aged 65 or older, again followed by vulnerable individuals. 

This strategy has the potential to reduce deaths and hospitalizations by relative 35–40% with non-sterilizing vaccines and by 50–60% with sterilizing vaccines. Whereas the effect of non-sterilizing vaccines on newly confirmed cases is marginal, the relative reduction of confirmed cases of sterilizing vaccines may range from 25% to 30%.

Figure 2 displays the impact of these stepwise strategies for non-sterilizing and sterilizing vaccines in terms of the relative reductions of deaths, hospitalizations, and new confirmed cases of COVID-19 infections in comparison to no vaccination.

The optimization algorithm yielded the same optimal sequences of population group to be vaccinated, when maximizing relative reduction in deaths or hospitalizations was considered in the targeting function. The magnitude in terms of relative reductions for the optimized strategies was similar for overall hospitalizations and hospitalizations requiring intensive care. Stepwise results of the sequential optimization when maximizing relative reduction in deaths are shown in the Appendix A (Figure A1, Figure A2, Figure A3 and Figure A4). The sequential optimization identified a non-sterilizing vaccine with 100% effectiveness in the sensitivity analysis, the same optimal sequence of prioritizing elderly, and vulnerable individuals to minimize COVID-19-related deaths and hospitalizations.

## 4. Discussion

### 4.1. Key Findings Base-Case Analysis and System-Relevant Target Group Considerations

If the goal is to minimize COVID-19-related deaths, hospitalizations, or both, elderly and vulnerable persons should be prioritized for vaccination, followed by middle-aged persons, healthcare workers, and younger individuals. Although prioritizing healthcare workers in the initial phase may result in smaller initial relative reductions in COVID-19-related deaths and hospitalizations compared with prioritizing the elderly or vulnerable persons, additional ethical and system-relevant implications should added in this decision [10,16,33]. For example, frontline HCW have a particularly high work-related risk of infection when following their professional responsibilities, in order to maintain the health system and serve the society. Therefore, the protection of HCW may receive a high priority in order to maximize their occupational safety and to ensure “risk-compensatory” justice in a situation where they are exposed to a higher risk. In addition, following the principle of instrumental and social relevance, the preservation of a functioning healthcare system due to the vaccination of healthcare workers (and other essential workers) must be considered. Despite all protection measures, healthcare workers may still play a multiplier role in the spread of the virus that is not fully captured in the heterogeneous contact pattern in our model. A vaccine providing sterilizing immunity to healthcare staff indirectly protects others including other patients, residents of nursing homes, and vulnerable groups [34]. This aspect follows the principle of utility maximization, but it has already been (partially) taken into account in our modeling results, as the model included contacts of healthcare staff within healthcare institutions such as nursing homes. In addition, knowing that healthcare workers are vaccinated may avoid patients’ delaying of important medical consultation and check-ups or hospital visits or preventive examinations due to patients’ fear of becoming infected by HCW. Finally, logistical aspects and efficiency in supply and distribution must be considered in prioritization decisions. For example, if a mobile vaccination team visits a nursing home to vaccinate its residents, it may be most efficient that the healthcare staff of this nursing home is vaccinated during the same visit to speed up the overall vaccination process, to avoid unnecessary startup costs, and to avoid loss of vaccine doses due to interruption of the delivery cold chain.

### 4.2. Stepwise Optimization Vaccinating 2.5 Million Individuals and the Role of NPIs

Our stepwise optimization focusing on avoided deaths leads to the same sequence of prioritization as optimization focusing on avoided hospitalizations; that is, elderly and vulnerable persons should be prioritized for vaccination. Assuming availability of a non-sterilizing vaccine for approximately 2.5 million individuals, vaccinating the elderly followed by vulnerable persons avoids approximately one third of deaths and hospitalizations when compared to no vaccination over a 6 month time horizon. In this phase, prioritizing vaccination of healthcare workers yields comparable results, due to the relatively moderate fraction of healthcare workers in the total population. While non-sterilizing and sterilizing vaccines led to the same prioritization focusing on the highest relative reduction in death and hospitalizations. In the Austrian context, our modeling results support currently pursued vaccination strategies [34]. However, modeling the use of sterilizing vaccines shows effective and sustained reductions in hospitalizations and deaths and, in particular, greater reductions in the spread of infections, which were expected.

It must be emphasized that these results assume that other protective measures against infectious events (e.g., face masks, distancing, and hygiene) are still maintained. If the goal is to reduce new infections, different prioritization sequences may be favored, including those giving higher priority to healthcare workers and (younger) people with more work or social contacts and mobility. This should be considered in the later phases of the vaccination strategy. We use our agent-based COVID-19 model with the stepwise optimization algorithm and updated model parameters to provide evidence for such phases. 

### 4.3. Link to Health Policy Decision Making and Further Strengths of the Study

To our knowledge, our study launched in summer 2020 is the first published modeling study on COVID-19 vaccination distribution using an agent-based simulation and a model framework that was directly informed by healthcare decision making bodies. Our Standing Policy and Expert Panel guided the project in defining upfront relevant optimization criteria in the Austrian context, potential target group categories, and further assumptions including participation rates as well as interpretations and communication of results. The aim of the study was to inform health policy decision makers and authorities, in a manner, that strategies related to the distribution of COVID-19 vaccines are evidence-based. Therefore, the members of the expert panel were informed about preliminary result as soon as of 17 November 2020, and results were communicated to the respective committees and authorities to be considered when developing and announcing the Austrian vaccination strategy. As previously mentioned, the implemented vaccination strategy in Austria is in line with our research results. In Austria, at the beginning, the vaccination plan included home residents, clinical staff, the over 85 year olds, and then the target group of the over 65 year olds. In mid-April 2021, the new vaccination phase can already begin in some federal states with the preparation for the vaccinations for and the invitations to under 65 year olds.

The agent-based modeling approach provides the most flexibility in incorporating details such as contact networks, nursing home facilities, distribution of age specific comorbidities, household structures, or impact of non-pharmacological prevention strategies and contact-tracing measures as compared to applying mathematical differential equation models [27,35]. Therefore, our model is noteworthy. Firstly, its detailed spatial and demographic image of the entire Austrian population is unique. Secondly, a detailed underlying contact network based on locations including households, nursing homes, workplaces, and schools is considered. Thirdly, it includes the ability for tracing of agent–agent contacts and vaccinating specific subpopulations. It is important to note that our study would not have been possible if we could not have built on prior work, particularly the development of a Generic Population Concept for Austria (GEPOC) [22] within the ‘Decision Support for Health Policy and Planning: Methods, Models and Technologies based on Existing Health Care Data’ (DEXHELPP) project, which was part of the ‘COMET—Competence Centers for Excellent Technologies’ funded by the Austrian government and organized by the Austrian Research Promotion Agency (FFG) [36].

### 4.4. Limitations of the Study

As all modeling studies, our study has several major limitations. First, the analyses generally do not account for the negative effects of COVID-19-related HCW absenteeism on the outcomes presented. Second, contact behavior and the likelihood of infection of HCW were modeled according to the general population, which may not fully capture specific infection risk in healthcare settings. Specifically, the risk may be increased when HCW are being exposed to patients who might be infected with SARS-CoV2, such as in primary care practice or emergency wards. However, the 3.4-fold increase in infection risk of HCW, assumed by Moore et al. [37] in a vaccination study for the UK, could not be confirmed by Austrian data. In Austria, a stepwise implemented use of personal protection equipment was introduced, and hygiene concepts were implemented early in the pandemic. Third, for vulnerable groups, age-specific distributions of risk factors were partially taken from German surveys, which however, likely do not differ substantially from the Austrian context. Fourth, contact behavior of vulnerable groups was modeled according to the general population. Studies on changes in interpersonal contacts during the COVID-19 epidemic are ongoing (update POLYMOD [17]). Fifth, we did not consider logistics and time required for vaccination; that is, we assumed individuals are vaccinated more or less at the same time. Although this simplification has no major impact on the selection of the prioritization strategies, the level of effect of the vaccination differs when accounting for substantial delays in vaccine delivery. Sixth, we applied a conservative assumption for vaccination effectiveness of 70% in the general population. Currently approved vaccines showed greater effectiveness in initial trials [38,39]. Consequently, the expected effects of vaccination on defined population level outcomes is likely even higher. However, our sensitivity analyses across vaccination effectiveness ranges, including currently available estimates, showed that the prioritization sequence of individuals does not change with effectiveness between 70% and 100%, covering the range of currently available vaccines. Further vaccines based on various technologies are still in development and effectiveness of these upcoming vaccines is not yet known. It is, however, likely that only vaccines with a lower bound or still substantial effectiveness are approved and/or used going forward. In the future, we may also have evidence of subgroup-specific vaccine efficacy in risk groups vulnerable to severe outcomes, vaccine efficacy in children, and the relative capability to prevent transmission of SARS-CoV-2 or complete sterilizing immunity. In addition, the unpredictable effects of emerging SARS-CoV2 mutations on infection rate, incidence of symptoms, mortality, and, specifically, the effect of so-called immune escape variants on vaccine efficacy in terms of preventing either infections in general or severe infections and hospitalizations cannot yet be predicted at the moment. These vaccine characteristics will be included on updated analyses as soon as such data become available. 

### 4.5. Comparison of Results with Findings from Other Published Work

Our results are in line with other modeling studies, although comparability to other simulation studies is limited. Differences in modeling studies include the modeling approach, assumed reproduction rates (R0), assumed sterilizing nature of vaccines, and country specific assumptions including demographics, distributions of comorbidities, or contact pattern. For example, Moore et al. [37] showed “When structuring by age alone, the most efficacious reduction was found through an oldest-first approach“. They found less consistency in the optimal position of vaccinating comorbidities in the priority order, which varied between just before and just after the 60–80 age group depending on the sterilizing nature of vaccine. However, in a different approach to our study, they assumed an increased risk of death conditional on various comorbidities. They also modeled scenarios for vaccines that reduced the probabilities of becoming symptomatic or experiencing severe symptoms separately and assumed a high reproduction rate of R0 = 1.8. Bubar et al. [40] and Matrajt et al. [41] only stratified by age, not explicitly accounting for comorbidities. Bubar et al. [40] found that across countries, individuals 60 years and older should be prioritized to minimize deaths. They identified two scenarios where vaccinating all adults or adults aged 20–49 would provide greater mortality benefits: “simultaneous conditions of transmission-mitigating behavior (R0 = 1.3), vaccine efficacy 80% or below, and 90% or higher transmission blocking” and “vaccines with very low efficacy in older adults, very high efficacy in younger adults, and declines in efficacy starting at 49 or 59, for a leaky vaccine” [40]. Matrajt et al. [41] considered a sterilizing vaccine and concluded that for “higher vaccine effectiveness, there is a switch to allocate vaccine to high-transmission (younger) age groups first for high vaccination coverage.” However, in these scenarios, an almost 100% participation in these high-transmission (younger) age groups was considered, which may be unlikely in reality. For low vaccination coverage, similar to our scenario of vaccinating 2.5 million individuals, vaccinating the elderly was preferred in their analysis. The study of Foy et al. also supports our results. Authors came to the conclusion that older age groups should be prioritized first. The prioritization of older age groups led to the greatest relative reduction in death regardless of vaccine efficacy [42]. 

### 4.6. Ethical Considerations

The results of our study should be interpreted and discussed considering further important social and ethical aspects and frameworks including self-determination and voluntariness, the non-compensation principle, justice, and equality of rights (“equal treatment for all”), medical-epidemiological overall benefit, system maintenance (state functions, healthcare, and public life), urgency (e.g., age, vulnerability, and social situation), and constitutional and legal conformity [16,33]. From an ethical point of view, medical and healthcare personnel require special prioritization, which is supported by various considerations, among them their exceptionally important function during the pandemic, since the possibility of healthcare capacity bottlenecks is one of the particular dangers of the COVID-19 pandemic. Specific legal approaches to prioritization also arise from the Austrian Epidemics Act 1950 (EpidemieG), as mandatory protective vaccinations can be ordered by public authorities, especially for members of healthcare professions.

### 4.7. Future Research Directions

In the future, our model can be applied to consider further subgroups with respect to social and demographic determinants, the timing of vaccination, differing vaccine effectiveness levels from different manufacturers or potentially different effectiveness of vaccines caused by COVID-19 mutations, and updated information on delivery of new doses of vaccines. With the advice of our Standing Policy and Expert Panel, we are currently using our model for further analysis in order to provide evidence for a continuing guide for policy decisions makers and public health authorities in later phases of the vaccination program based on updated model parameters.

## 5. Conclusions

Our decision-analytic study based on agent-based dynamic transmission simulation shows that, in order to minimize COVID-19-related deaths and hospitalizations, elderly and vulnerable persons should be prioritized for vaccination independent of the fact that the vaccine induces sterilizing or non-sterilizing immunity. Vaccinating 2.5 million elderly and vulnerable individuals, which accounts for approximately 30% of the Austrian population, with a non-sterilizing vaccine avoids approximately one third of hospitalizations and deaths compared with no vaccination in Austria. Related to the entire Austrian population, prioritizing vaccination of the relatively small group of healthcare workers yields comparable absolute results when elderly and vulnerable persons are vaccinated next. All of the results assume that other protective measures against the spread of SARS-CoV-2 are maintained. Further analyses are required to determine an optimal vaccination sequence for further outcomes, subgroups, updated data on the effectiveness of vaccines, the behavior of emerging mutations, and the participation rate in vaccination programs, considering explicit tradeoffs between health. social, economic, and ethical outcomes.

## Figures and Tables

**Figure 1 vaccines-09-00434-f001:**
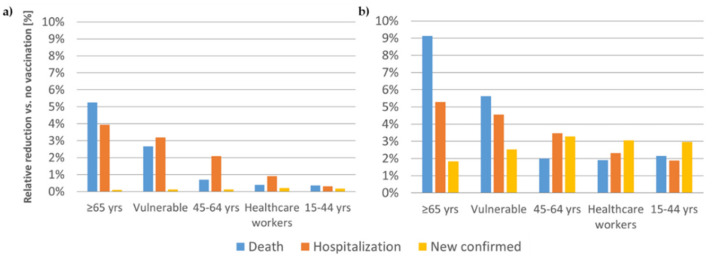
(**a**) Non-sterilizing vaccine: total impact of vaccinating the first 200,000 individuals in different target populations with a non-sterilizing vaccine on deaths, hospitalizations, and new confirmed infections. (**b**) Sterilizing vaccine: total impact of vaccinating the first 200,000 individuals in different target populations with a sterilizing vaccine on deaths, hospitalizations, and new confirmed infections. yrs—years; vulnerable: individuals with increased risk of severe COVID-19 disease once infected. Analysis assumed a vaccine effectiveness of 70% in the general population and of 60% in the age group 65+.

**Figure 2 vaccines-09-00434-f002:**
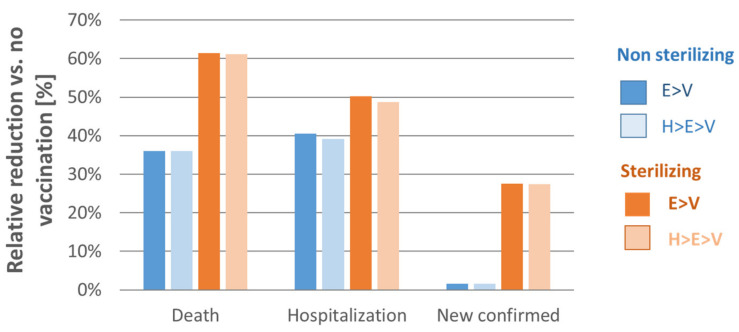
Total impact of vaccinating 2.45 million individuals on relative reductions in deaths, hospitalizations, and new confirmed infections. Crit.—critical, E—elderly, HCW—healthcare workers, Hosp.—hospitalizations, V—vulnerable. E > V: starting with the elderly (age 65+; 1.45 million), followed by vulnerable individuals (1 million) with increased risk of severe disease once infected. H > E > V: starting with healthcare workers (0.2 million), followed by the elderly (1.45 million) and then vulnerable individuals (0.8 million). Analysis assumed vaccine effectiveness to be 70% in the general population and 60% in the age group 65+. Optimal vaccination strategies were determined by maximizing relative reduction of death and hospitalizations.

## Data Availability

Further information on the underlying population model are provided in [17] (Paper September with model). Authors can be contacted for further information.

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
