# Peer review of "Targeted COVID-19 Vaccination (TAV-COVID) Considering Limited Vaccination Capacities—An Agent-Based Modeling Evaluation"

_vaccines, 2021, doi:10.3390/vaccines9050434_

Round 1
Reviewer 1 Report
The manuscript presented by John et al. describes a very interesting model for the decision-making of groups of individuals to be vaccinated in the context of COVID-19.
However, I would like to make some comments and considerations regarding the manuscript:
- I think the methodology section is very extensive. Thus, I would like to suggest to the authors a better match between the text presented in the methodology and the text presented in the discussion session;
- Figura 1: I suggest that figures 1a and 1b be formatted and presented in a single figure, figure 1;
- I think the tables presented in the appendix could be presented in the results section of the manuscript;
- The "discussion" section of the manuscript is very interesting and complete, but I suggest that the paragraphs are better presented, as they are very long. I think the text could be less dense and many long paragraphs could be fragments in the session.
Reviewer 2 Report
Dear Authors,
Thank you for your contribution. I recommend following suggestions to reconsider your manuscript. Please answer pointwise and incorporate all the changes, discussion, and suggested references in suitable places for us to be clear about the revised/edited manuscript. Thank you for your contribution.
SUGGESTIONS AS BELOW:
1.Line 126: Society for Medical Decision Making (SMDM) COVID-19 Decision Modeling Initiative
--Just curious abut this initiative- was this kind of initiative also during triaging for patients for distribution of ventilators etc.? I guess this is very challenging times and we need such protocol to prioritize the vaccines to the needy.
2.Line 134; decision-analytic modeling
- Was this protocol was used across the nation while distributing vaccines?
3.Line 175: In all simulations, we conservatively assumed a vaccination effectiveness of 70% for
individuals younger than 65 years and a reduced effectiveness of 60% for individuals 65 176
years of age and older
- I am little confused about the vaccine efficacy. From where we got this data. I was aware of 90% and above, please clarify. Was that calculated from the author’s study? If yes then what is the cause of the
- Similar study done recently, please consider discussing the same.
https://pubmed.ncbi.nlm.nih.gov/33388436/
- Line 81 Dependent on these characteristics, the vaccination strategy may focus on the protection of those who are at highest risk for severe disease, or the strategy could target on those who are mostly spreading the disease, or it could consider both aspects.
- This can be expanded “who are at highest risk for sever disease” like patient with cancer, HIV, pregnancy, immunosuppression. Consider following reference to be added:
https://www.ncbi.nlm.nih.gov/pmc/articles/PMC7151275/
https://pubmed.ncbi.nlm.nih.gov/33808066/
https://pubmed.ncbi.nlm.nih.gov/33773923/
Reviewer 3 Report
In this manuscript, Siebert et al. used an agent-based simulation model to evaluate the effect of different SARS-CoV-2 vaccination priorities and strategies in Austria. The aim is to provide guidance to the authorities to minimize the COVID-19 related deaths and hospitalizations. In their model, the total population represented by nine million Austrians is defined to five subgroups, including elderly (≥ 65 years), middle aged (45-64 years), younger (15-44 years), vulnerable, and health care workers (HCW).
Due to the onset of limited vaccine supply, the samples were first taken from an initially available vaccine batch of 200,000 individuals. The data showed that starting vaccination with elderly and vulnerable followed by middle-aged, HCW, and younger individuals can achieve maximum reduction of hospitalizations and deaths compared with no vaccination. After evaluation of fully vaccinating 200,000 individuals, authors further conducted a stepwise optimization for a larger sample size of 2.45 million individuals. Two independent optimization analyses were performed. The primary analysis was to minimize deaths and the secondary analysis was to minimize hospitalization. They found that 2.45 million individuals also yielded the same prioritization and avoided about one third of deaths and hospitalizations. However, stating vaccination with HCW led to slightly smaller reductions but maximizes occupational safety.
Overall, the text is well prepared and reveals important information about how to prioritize the vaccination before vaccines are fully available. Quality and importance of this manuscript reach the standard of Vaccines for its publication. However, some minor points have to be clarified before it is accepted.
âť… Population numbers of each subgroup, including elderly (≥ 65 years), middle aged (45-64 years), younger (15-44 years), vulnerable, and health care workers (HCW), for 200,000 and 2.45 million individuals should be provided in the text or figures. Their distribution will then be easily caught by readers.
âť… The definition of vulnerable may need to be more specific as the definition of elderly, middle aged, and younger are clearly based on ages. As for the health care workers, they are defined on the basis of occupation.
âť… Appendix A is unclear, in particular Figures 1A, 2A, 3A, and 4A. They need to be improved and revised.
